# Consistency of Spectral Partitioning of Uniform Hypergraphs under Planted Partition Model

**Debarghya Ghoshdastidar**    **Ambedkar Dukkipati**
Department of Computer Science & Automation
Indian Institute of Science
Bangalore – 560012, India
{debarghya.g,ad}@csa.iisc.ernet.in

## Abstract

Spectral graph partitioning methods have received significant attention from both practitioners and theorists in computer science. Some notable studies have been carried out regarding the behavior of these methods for infinitely large sample size (von Luxburg *et al.*, 2008; Rohe *et al.*, 2011), which provide sufficient confidence to practitioners about the effectiveness of these methods. On the other hand, recent developments in computer vision have led to a plethora of applications, where the model deals with multi-way affinity relations and can be posed as uniform hypergraphs. In this paper, we view these models as random $m$-uniform hypergraphs and establish the consistency of spectral algorithm in this general setting. We develop a planted partition model or stochastic blockmodel for such problems using higher order tensors, present a spectral technique suited for the purpose and study its large sample behavior. The analysis reveals that the algorithm is consistent for $m$-uniform hypergraphs for larger values of $m$, and also the rate of convergence improves for increasing $m$. Our result provides the first theoretical evidence that establishes the importance of $m$-way affinities.

## 1  Introduction

The central theme in approaches like kernel machines [1] and spectral clustering [2, 3] is the use of symmetric matrices that encode certain similarity relations between pairs of data instances. This allows one to use the tools of matrix theory to design efficient algorithms and provide theoretical analysis for the same. Spectral graph theory [4] provides classic examples of this methodology, where various hard combinatorial problems pertaining to graphs are relaxed to problems of matrix theory. In this work, we focus on spectral partitioning, where the aim is to group the nodes of a graph into disjoint sets using the eigenvectors of the adjacency matrix or the Laplacian operator. A statistical framework for this partitioning problem is the planted partition or stochastic blockmodel [5]. Here, one assumes the existence of an unknown map that partitions the nodes of a random graph, and the probability of occurrence of any edge follows the partition rule. In a recent work, Rohe *et al.* [6] studied normalized spectral clustering under the stochastic blockmodel and proved that, for this method, the fractional number of misclustered nodes goes to zero as the sample size grows.

However, recent developments in signal processing, computer vision and statistical modeling have posed numerous problems, where one is interested in computing multi-way similarity functions that compute similarity among more than two data points. A few applications are listed below.
**Example 1.**  In geometric grouping, one is required to cluster points sampled from a number of geometric objects or manifolds [7]. Usually, these objects are highly overlapping, and one cannot use standard distance based pairwise affinities to retrieve the desired clusters. Hence, one needs to construct multi-point similarities based on the geometric structure. A special case is the subspace clustering problem encountered in motion segmentation [7], face clustering [8] etc.

**Example 2.** The problem of point-set matching [9] underlies several problems in computer vision including image registration, object recognition, feature tracking etc. The problem is often formulated as finding a strongly connected component in a uniform hypergraph [9, 10], where the strongly connected component represents the correct matching. This formulation has the flavor of the standard problem of detecting cliques in random graphs.

Both of the above problems are variants of the classic hypergraph partitioning problem, that arose in the VLSI community [11] in 1980s, and has been an active area of research till date [12]. Spectral approaches for hypergraph partitioning also exist in the literature [13, 14, 15], and various definitions of the hypergraph Laplacian matrix has been proposed based on different criteria. Recent studies [16] suggest an alternative representation of uniform hypergraphs in terms of the "affinity tensor". Tensors have been popular in machine learning and signal processing for a considerable time (see [17]), and have even found use in graph partitioning and detecting planted partitions [17, 18]. But their role in hypergraph partitioning have been mostly overlooked in the literature. Recently, techniques have emerged in computer vision that use such affinity tensors in hypergraph partitioning [8, 9].

This paper provides the first consistency result on uniform hypergraph partitioning by analyzing the spectral decomposition of the affinity tensor. The main contributions of this work are the following.
**(1)** We propose a planted partition model for random uniform hypergraphs similar to that of graphs [5]. We show that the above examples are special cases of the proposed partition model.
**(2)** We present a spectral technique to extract the underlying partitions of the model. This method relies on a spectral decomposition of tensors [19] that can be computed in polynomial time, and hence, it is computationally efficient than the tensorial approaches in [10, 8].
**(3)** We analyze the proposed approach and provide almost sure bounds on the number of misclustered nodes. Our analysis reveals that the presented method is consistent almost surely in the grouping problem and for detection of a strongly connected component, whenever one uses $m$-way affinities for any $m \geq 3$ and $m \geq 4$, respectively. The derived rate of convergence also shows that the use of higher order affinities lead to a faster decay in the number of misclustered nodes.
**(4)** We numerically demonstrate the performance of the approach on benchmark datasets.

## 2 Planted partitions in random uniform hypergraphs

We describe the planted partition model for an undirected unweighted graph. Let $\psi : \{1, \ldots, n\} \to \{1, \ldots, k\}$ be an (unknown) partition of $n$ nodes into $k$ disjoint groups, *i.e.*, $\psi_i = \psi(i)$ denotes the partition in which node-$i$ belongs. We also define an assignment matrix $Z_n \in \{0, 1\}^{n \times k}$ such that $(Z_n)_{ij} = 1$ if $j = \psi_i$, and 0 otherwise. For some unknown symmetric matrix $B \in [0, 1]^{k \times k}$, the random graph on the $n$ nodes contains the edge $(i, j)$ with probability $B_{\psi_i \psi_j}$. Let the symmetric matrix $A_n \in \{0, 1\}^{n \times n}$ be a realization of the affinity matrix of the random graph on $n$ nodes. The aim is to identify $Z_n$ given the matrix $A_n$. In some cases, one also needs to estimate the entries in $B$. One can hope to achieve this goal for the following reason: If $\mathcal{A}_n \in \mathbb{R}^{n \times n}$ contains the expected values of the entries in $A_n$ conditioned on $B$ and $\psi$, then one can write $\mathcal{A}_n$ as $\mathcal{A}_n = Z_n B Z_n^T$ [6]. Thus, if one can find $\mathcal{A}_n$, then this relation can be used to find $Z_n$.

We generalize the partition model to uniform hypergraphs. A hypergraph is a structure on $n$ nodes with multi-way connections or hyperedges. Formally, each hyperedge in an undirected unweighted hypergraph is a collection of an arbitrary number of vertices. A special case is that of $m$-uniform hypergraph, where each hyperedge contains exactly $m$ nodes. One can note that a graph is a 2-uniform hypergraph. An often cited example of uniform hypergraph is as follows [10]. Let the nodes be representative of points in an Euclidean space, where a hyperedge exists if the points are collinear. For $m = 2$, we obtain a complete graph that does not convey enough information about the nodes. However, for $m = 3$, the constructed hypergraph is a union of several connected components, each component representing a set of collinear points. The affinity relations of an $m$-uniform hypergraph can be represented in the form of an $m^{th}$-order tensor $A_n \in \{0, 1\}^{n \times n \times \ldots \times n}$, which we call an affinity tensor. The entry $(A_n)_{i_1 \ldots i_m} = 1$ if there exists a hyperedge on nodes $i_1, \ldots, i_m$. One can observe that the tensor is symmetric, *i.e.*, invariant under any permutation of indices. In some works [16], the tensor is scaled by a factor of $1/(m-1)!$ for certain reasons.

Let $\psi$ and $Z_n$ be as defined above, and $B \in [0, 1]^{k \times \ldots \times k}$ be an $m^{th}$-order $k$-dimensional symmetric tensor. The random $m$-uniform hypergraph on the $n$ nodes is constructed such that a hyperedge occurs on nodes $i_1, \ldots, i_m$ with probability $B_{\psi_{i_1} \ldots \psi_{i_m}}$. If $A_n$ is a random affinity tensor of the

hypergraph, our aim is to find $Z_n$ or $\psi$ from $A_n$. Notice that if $\mathcal{A}_n \in \mathbb{R}^{n \times \cdots \times n}$ contains the expected values of the entries in $A_n$, then one can write the entries in $\mathcal{A}_n$ as

$$(\mathcal{A}_n)_{i_1 \ldots i_m} = B_{\psi_{i_1} \ldots \psi_{i_m}} = \sum_{j_1, \ldots, j_m = 1}^{k} B_{j_1 \ldots j_m} (Z_n)_{i_1 j_1} \ldots (Z_n)_{i_m j_m}. \qquad (1)$$

The subscript $n$ in the above terms emphasizes their dependence on the number of nodes. We now describe how two standard applications in computer vision can be formulated as the problem of detecting planted partitions in uniform hypergraphs.

## 2.1 Subspace clustering problem

In motion segmentation [7, 20] or illumination invariant face clustering [8], the data belong to a high dimensional space. However, the instances belonging to each cluster approximately span a low-dimensional subspace (usually, of dimension 3 or 4). Here, one needs to check whether $m$ points approximate such a subspace, where this information is useful only when $m$ is larger than the dimension of the underlying subspace of interest. The model can be represented as an $m$-uniform hypergraph, where a hyperedge occurs on $m$ nodes whenever they approximately span a subspace.

The partition model for this problem is similar to the standard four parameter blockmodel [6]. The number of partitions is $k$, and each partition contains $s$ nodes, *i.e.*, $n = ks$. There exists probabilities $p \in (0, 1]$ and $q \in [0, p)$ such that any set of $m$ vectors span a subspace with probability $p$ if all $m$ vectors belong to the same group, and with probability $q$ if they come from different groups. Thus, the tensor $B$ has the form $B_{i \ldots i} = p$ for all $i = 1, \ldots, k$, and $B_{i_1 \ldots i_m} = q$ for all the other entries.

## 2.2 Point set matching problem

We consider a simplified version of the matching problem [10], where one is given two sets of points of interest, each of size $s$. In practice, these points may come from two different images of the same object or scene, and the goal is to match the corresponding points. One can see that there are $s^2$ candidate matches. However, if one considers $m$ correct matches then certain properties are preserved. For instance, let $i_1, \ldots, i_m$ be some points from the first image, and $i_1', \ldots, i_m'$ be the corresponding points in the second image, then the angles or ratio of areas of triangles formed among these points are more or less preserved [9]. Thus, the set of matches $(i_1, i_1'), \ldots, (i_m, i_m')$ have a certain connection, which is usually not present if the matches are not exact.

The above model is an $m$-uniform hypergraph on $n = s^2$ nodes, each node representing a candidate match, and a hyperedge is formed if properties (like preservation of angles) is satisfied by $m$ candidate matches. Here, one can see that there are only $s = \sqrt{n}$ correct matches, which have a large number of hyperedges among them, whereas very few hyperedges may be present for other combinations. Thus, the partition model has two groups of size $\sqrt{n}$ and $(n - \sqrt{n})$, respectively. For $p, q \in [0, 1]$, $p \gg q$, $p$ denotes the probability of a hyperedge among $m$ correct matches and for any other $m$ candidates, there is a hyperedge with probability $q$. Thus, if the first partition is the strongly connected component, then we have $B \in [0, 1]^{2 \times \cdots \times 2}$ with $B_{1 \ldots 1} = p$ and $B_{i_1 \ldots i_m} = q$ otherwise.

## 3 Spectral partitioning algorithm and its consistency

Before presenting the algorithm, we provide some background on spectral decomposition of tensors. In the related literature, one can find a number of significantly different characterizations of the spectral properties of tensors. While the work in [16] builds on a variational characterization, De Lathauwer *et al.* [19] provide an explicit decomposition of a tensor in the spirit of the singular value decomposition of matrices. The second line of study is more appropriate for our work since our analysis significantly relies on the use of Davis-Kahan perturbation theorem [21] that uses an explicit decomposition, and has been often used to analyze spectral clustering [2, 6].

The work in [19] provides a way of expressing any $m^{th}$-order $n$-dimensional symmetric tensor, $A_n$, as a mode-$k$ product [19] of a certain core tensor with $m$ orthonormal matrices, where each orthonormal matrix is formed from the orthonormal left singular vectors of $\widehat{A}_n \in \{0, 1\}^{n \times n^{m-1}}$,

whose entries, for all $i = 1, \ldots, n$ and $j = 1, \ldots, n^{m-1}$, are defined as

$$(\widehat{A}_n)_{ij} = (A_n)_{i_1 i_2 \ldots i_m}, \qquad \text{if } i = i_1 \text{ and } j = 1 + \sum_{l=2}^{m} (i_l - 1) n^{l-2}. \tag{2}$$

The above matrix $\widehat{A}_n$, often called the mode-1 flattened matrix, forms a key component of the partitioning algorithm. Later, we show that the leading $k$ left singular vectors of $\widehat{A}_n$ contain information about the true partitions in the hypergraph. It is easier to work with the symmetric matrix $W_n = \widehat{A}_n \widehat{A}_n^T \in \mathbb{R}^{n \times n}$, whose eigenvectors correspond to the left singular vectors of $\widehat{A}_n$. The spectral partitioning algorithm is presented in Algorithm 1, which is quite similar to the normalized spectral clustering [2]. Such a tensor based approach was first studied in [7] for geometric grouping. Subsequent improvements of the algorithm were proposed in [22, 20]. However, we deviate from these methods as we do not normalize the rows of the eigenvector matrix. The method in [9] also uses the largest eigenvector of the flattened matrix for the point set matching problem. This is computed via tensor power iterations. To keep the analysis simple, we do not use such iterations. The complexity of Algorithm 1 is $O(n^{m+1})$, which can be significantly improved using sampling techniques as in [7, 9, 20]. The matrix $D_n$ is used for normalization as in spectral clustering.

---

**Algorithm 1** Spectral partitioning of $m$-uniform hypergraph

1. From the $m^{th}$-order affinity tensor $A_n$, construct $\widehat{A}_n$ using (2).
2. Let $W_n = \widehat{A}_n \widehat{A}_n^T$, and $D_n \in \mathbb{R}^{n \times n}$ be diagonal with $(D_n)_{ii} = \sum_{j=1}^{n} (W_n)_{ij}$.
3. Set $L_n = D_n^{-1/2} W_n D_n^{-1/2}$.
4. Compute leading $k$ orthonormal eigenvectors of $L_n$, denoted by matrix $X_n \in \mathbb{R}^{n \times k}$.
5. Cluster the rows of $X_n$ into $k$ clusters using $k$-means clustering.
6. Assign node-$i$ of hypergraph to $j^{th}$ partition if $i^{th}$ row of $X_n$ is grouped in $j^{th}$ cluster.

---

An alternative technique of using eigenvectors of Laplacian matrix is often preferred in graph partitioning [3], and has been extended to hypergraphs [13, 15]. Unlike the flattened matrix, $\widehat{A}_n$, in Algorithm 1, such Laplacians do not preserve the spectral properties of a higher-order structure such as the affinity tensor that accurately represents the affinities of the hypergraph. Hence, we avoid the use of hypergraph Laplacian.

### 3.1 Consistency of above algorithm

We now comment on the error incurred by Algorithm 1. For this, let $M_n$ be the set of nodes that are incorrectly clustered by Algorithm 1. It is tricky to formalize the definition of $M_n$ in clustering problems. We follow the definition of $M_n$ given in [6] that requires some details of the analysis and hence, a formal definition is postponed till Section 4. In addition, we need the following terms. The analysis depends on the tensor $B \in [0,1]^{k \times \cdots \times k}$ of the underlying random model. Let $\widehat{B} \in [0,1]^{k \times k^{m-1}}$ be the flattening of tensor $B$ using (2). We also define a matrix $C_n \in \mathbb{R}^{k \times k}$ as

$$C_n = (Z_n^T Z_n)^{1/2} \widehat{B} (Z_n^T Z_n)^{\otimes(m-1)} \widehat{B}^T (Z_n^T Z_n)^{1/2}, \tag{3}$$

where $(Z_n^T Z_n)^{\otimes(m-1)}$ is the $(m-1)$-times Kronecker product of $Z_n^T Z_n$ with itself. Use of such Kronecker product is quite common in tensor decompositions (see [19]). Observe that the positive semi-definite matrix $C_n$ contains information regarding the connectivity of clusters (stored in $B$) and the cluster sizes (diagonal entries of $Z_n^T Z_n$). Let $\lambda_k(C_n)$ be the smallest eigenvalue of $C_n$, which is non-negative. In addition, define $\mathcal{D}_n \in \mathbb{R}^{n \times n}$ as the expectation of the diagonal matrix $D_n$. One can see that $(\mathcal{D}_n)_{ii} \leq n^m$ for all $i = 1, \ldots, n$. Let $\underline{\mathcal{D}}_n$ and $\overline{\mathcal{D}}_n$ be the smallest and largest values in $\mathcal{D}_n$. Also, let $\underline{S}_n$ and $\overline{S}_n$ be the sizes of the smallest and largest partitions, respectively. We have the following bound on the number of misclustered nodes.

**Theorem 1.** *If there exists $N$ such that for all $n > N$,*

$$\delta_n := \left( \frac{\lambda_k(C_n)}{\overline{\mathcal{D}}_n} - \frac{2n^{m-1}}{\underline{\mathcal{D}}_n} \right) > 0 \qquad and \qquad \underline{\mathcal{D}}_n \geq n^m (m-1)! \sqrt{\frac{2}{\log n}},$$

*and if* $(\log n)^{3/2} = o\left(\delta_n n^{\frac{m-1}{2}}\right)$, *then the number of misclustered nodes*

$$|M_n| = O\left(\frac{\overline{S}_n(\log n)^2 n^{m+1}}{\delta_n^2 \underline{\mathcal{D}}_n^2}\right) \quad \textit{almost surely.}$$

The above result is too general to provide conclusive remarks about consistency of the algorithm. Hence, we focus on two examples, precisely the ones described in Sections 2.1 and 2.2. However, without loss of generality, we assume here that $q > 0$ since otherwise, the problem of detecting the partitions is trivial (at least for reasonably large $n$) as we can construct the partitions only based on the presence of hyperedges. The following results are proved in the appendix. The proofs mainly depend on computation of $\lambda_k(C_n)$, which can be derived for the first example, while for the second, it is enough to work with a lower bound of $\lambda_k(C_n)$. Further, in the first example, we make the result general by allowing the number of clusters, $k$, to grow with $n$ under certain conditions.

**Corollary 2.** *Consider the setting of subspace clustering described in Section 2.1. If the number of clusters $k$ satisfy $k = O\left(n^{\frac{1}{2m}}(\log n)^{-1}\right)$, then the conditions in Theorem 1 are satisfied and* $|M_n| = O\left(\frac{k^{2m-1}(\log n)^2}{n^{m-2}}\right) = O\left(\frac{(\log n)^{3-2m}}{n^{m-3+\frac{1}{2m}}}\right)$ *almost surely. Hence, for $m > 2$, $|M_n| \to 0$ a.s. as $n \to \infty$, i.e., the algorithm is consistent. For $m = 2$, we can only conclude $\frac{|M_n|}{n} \to 0$ a.s.*

From the above result, it is evident that the rate of convergence improves as $m$ increases, indicating that, ignoring practical considerations, one should prefer the use of higher order affinities. However, the condition of number of clusters becomes more strict in such cases. We note here that our result and conditions are quite similar to those given in [6] for the case of four-parameter block-model. Thus, Algorithm 1 is comparable to spectral clustering [6]. Next, we consider the setting of Section 2.2.

**Corollary 3.** *For the problem of point set matching described in Section 2.2, the conditions in Theorem 1 are satisfied for $m \geq 3$ and $|M_n| = O\left(\frac{(\log n)^2}{n^{m-3}}\right)$ a.s. Hence, for $m > 3$, $|M_n| \to 0$ a.s. as $n \to \infty$, i.e., the algorithm is consistent. For $m = 3$, we can only conclude $\frac{|M_n|}{n} \to 0$ a.s.*

The above result shows, theoretically, why higher order matching provides high accuracy in practice [9]. It also suggests that increase in the order of tensor will lead to a better convergence rate. We note that the following result does not hold for graphs ($m = 2$). In Corollary 3, we used the fact that the smaller partition is of size $s = \sqrt{n}$. The result can be made more general in terms of $s$, *i.e.*, for $m > 4$, if $s \geq \frac{3p}{q^3}$ eventually, then Algorithm 1 is consistent.

Before providing the detailed analysis (proof of Theorem 1), we briefly comment on the model considered here. In Section 2, we have followed the lines of [6] to define the model with $\mathcal{A}_n = Z_n B Z_n^T$. However, this would mean that the diagonal entries in $\mathcal{A}_n$ are non-negative, and hence, there is a non-zero probability of formation of self loops that is not common in practice. The same issue exists for hypergraphs. To avoid this, one can add a correction term to $\mathcal{A}_n$ so that the entries with repeated indices become zero. Under this correction, conditions in Theorem 1 should not change significantly. This is easy to verify for graphs, but it is not straightforward for hypergraphs.

## 4 Analysis of partitioning algorithm

In this section, we prove Theorem 1. The result follows from a series of lemmas. The proof requires defining certain terms. Let $\widehat{\mathcal{A}_n}$ be the flattening of the tensor $\mathcal{A}_n$ defined in (1). Then we can write $\widehat{\mathcal{A}_n} = Z_n \widehat{B}(Z_n^T)^{\otimes(m-1)}$, where $(Z_n^T)^{\otimes(m-1)}$ is $(m-1)$-times Kronecker product of $Z_n^T$ with itself. Along with the definitions in Section 3, let $\mathcal{W}_n \in \mathbb{R}^{n \times n}$ be the expectation of $W_n$, and $\mathcal{L}_n = \mathcal{D}_n^{-1/2} \mathcal{W}_n \mathcal{D}_n^{-1/2}$. One can see that $\mathcal{W}_n$ can be written as $\mathcal{W}_n = \widehat{\mathcal{A}_n}\widehat{\mathcal{A}_n}^T + \mathcal{P}_n$, where $\mathcal{P}_n$ is a diagonal matrix defined in terms of the entries in $\widehat{\mathcal{A}_n}$. The proof contains the following steps:
**(1)** For any fixed $n$, we show that if $\delta_n > 0$ (stated in Theorem 1), the leading $k$ orthonormal

eigenvectors of $\mathcal{L}_n$ has $k$ distinct rows, where each row is a representative of a partition.

(2) Since, $\mathcal{L}_n$ is not the expectation of $L_n$, we derive a bound on the Frobenius norm of their difference. The bound holds almost surely for all $n$ if eventually $\underline{\mathcal{D}}_n \geq n^m(m-1)!\sqrt{\frac{2}{\log n}}$.

(3) We use a version of Davis-Kahan sin-$\Theta$ theorem given in [6] that almost surely bounds the difference in the leading eigenvectors of $L_n$ and $\mathcal{L}_n$ if $(\log n)^{3/2} = o\left(\delta_n n^{\frac{m-1}{2}}\right)$.

(4) Finally, we rely on [6, Lemma 3.2], which holds in our case, to define the set of misclustered nodes $M_n$, and its size is bounded almost surely using the previously derived bounds.

We now present the statements for the above results. The proofs can be found in the appendix.

**Lemma 4.** *Fix $n$ and let $\delta_n$ be as defined in Theorem 1. If $\delta_n > 0$, then there exists $\mu_n \in \mathbb{R}^{k \times k}$ such that the columns of $Z_n \mu_n$ are the leading $k$ orthonormal eigenvectors of $\mathcal{L}_n$. Moreover, for nodes $i$ and $j$, $\psi_i = \psi_j$ if and only if the $i^{th}$ and $j^{th}$ rows of $Z_n \mu_n$ are identical.*

Thus, clustering the rows of $Z_n \mu_n$ into $k$ clusters will provide the true partitions, and the cluster centers will precisely be these rows. The condition $\delta_n > 0$ is required to ensure that the eigenvalues corresponding to the columns of $Z_n \mu_n$ are strictly greater than other eigenvalues. The requirement of a positive eigen-gap is essential for analysis of any spectral partitioning method [2, 23]. Next, we focus on deriving the upper bound for $\|L_n - \mathcal{L}_n\|_F$.

**Lemma 5.** *If there exists $N$ such that $\underline{\mathcal{D}}_n \geq n^m(m-1)!\sqrt{\frac{2}{\log n}}$ for all $n > N$, then*

$$\|L_n - \mathcal{L}_n\|_F \leq \frac{4n^{\frac{m+1}{2}}\log n}{\underline{\mathcal{D}}_n} , \qquad \text{almost surely.} \tag{4}$$

The condition in the above result implies that each vertex is reasonably connected to other vertices of the hypergraph, *i.e.*, there are no outliers. It is easy to satisfy this condition in the stated examples as $\underline{\mathcal{D}}_n \geq q^2 n^m$ and hence, it holds for all $q > 0$. Under the condition, one can also see that the bound in (4) is $O\left(\frac{(\log n)^{3/2}}{n^{\frac{m-1}{2}}}\right)$ and hence goes to zero as $n$ increases. Note that in Lemma 4, $\delta_n > 0$ need not hold for all $n$, but if it holds eventually, then we can choose $N$ such that the conditions in Lemmas 4 and 5 both hold for all $n > N$. Under such a case, we use the Davis-Kahan perturbation theorem [21] as stated in [6, Theorem 2.1] to claim the following.

**Lemma 6.** *Let $X_n \in \mathbb{R}^{n \times k}$ contain the leading $k$ orthonormal eigenvectors of $L_n$. If $(\log n)^{3/2} = o\left(\delta_n n^{\frac{m-1}{2}}\right)$ and there exists $N$ such that $\delta_n > 0$ and $\underline{\mathcal{D}}_n \geq n^m(m-1)!\sqrt{\frac{2}{\log n}}$ for all $n > N$, then there exists an orthonormal (rotation) matrix $O_n \in \mathbb{R}^{k \times k}$ such that*

$$\|X_n - Z_n \mu_n O_n\|_F \leq \frac{16n^{\frac{m+1}{2}}\log n}{\delta_n \underline{\mathcal{D}}_n} , \qquad \text{almost surely.} \tag{5}$$

The condition $(\log n)^{3/2} = o\left(\delta_n n^{\frac{m-1}{2}}\right)$ is crucial as it ensures that the difference in eigenvalues of $L_n$ and $\mathcal{L}_n$ decays much faster than the eigen-gap in $\mathcal{L}_n$. This condition requires the eigen-gap (lower bounded by $\delta_n$) to decay at a relatively slow rate, and is necessary for using [6, Theorem 2.1]. The bound (5) only says that rows of $X_n$ converges to some rotation of the rows of $Z_n \mu_n$. However, this is not an issue since the $k$-means algorithm is expected to perform well as long as the rows of $X_n$ corresponding to each partition are tightly clustered, and the $k$ clusters are well-separated. Now, let $z_1, \ldots, z_n$ be the rows of $Z_n$, and let $c_i$ be the center of the cluster in which $i^{th}$ row of $X_n$ is grouped for each $i \in \{1, \ldots, n\}$. We use a key result from [6] that is applicable in our setting.

**Lemma 7.** *[6, Lemma 3.2] For the matrix $O_n$ from Lemma 6, if $\|c_i - z_i \mu_n O_n\|_2 < \frac{1}{\sqrt{2\overline{S}_n}}$, then $\|c_i - z_i \mu_n O_n\|_2 < \|c_i - z_j \mu_n O_n\|_2$ for all $z_j \neq z_i$.*

This result hints that one may use the definition of correct clustering as follows. Node-$i$ is correctly clustered if its center $c_i$ is closer to $z_i \mu_n O_n$ than the rows corresponding to other partitions. A sufficient condition to satisfy this definition is $\|c_i - z_i \mu_n O_n\|_2 < \frac{1}{\sqrt{2\overline{S}_n}}$. Hence, the set of misclustered nodes is defined as [6]

$$M_n = \left\{ i \in \{1, \ldots, n\} : \|c_i - z_i \mu_n O_n\|_2 \geq \frac{1}{\sqrt{2\overline{S}_n}} \right\}. \tag{6}$$

It is easy to see that if $M_n$ is empty, *i.e.*, all nodes satisfy the condition $\|c_i - z_i\mu_n O_n\|_2 < \frac{1}{\sqrt{2\overline{S}_n}}$, then the clustering leads to true partitions, and does not incur any error. Hence, for statements, where $|M_n|$ is small (at least compared to $n$), one can always use such a definition for misclustered nodes. The next result provides a simple bound on $|M_n|$, that immediately leads to Theorem 1.

**Lemma 8.** *If the $k$-means algorithm achieves its global optimum, then the set $M_n$ satisfies*

$$|M_n| \leq 8\overline{S}_n\|X_n - Z_n\mu_n O_n\|_F^2 . \tag{7}$$

In practice, $k$-means algorithm tries to find a local minimum, and hence, one should run this step with multiple initializations to achieve a global minimum. However, empirically we found that good performance is achieved even if we use a single run of $k$-means. From above lemma, it is straightforward to arrive at Theorem 1 by using the bound in Lemma 6.

## 5 Experiments

### 5.1 Validation of Corollaries 2 and 3

We demonstrate the claims of Corollaries 2 and 3, where we stated that for higher order tensors, the number of misclustered nodes decays to zero at a faster rate. We run Algorithm 1 on both the models of subspace clustering and point-set matching, varying the number of nodes $n$, the results for each $n$ being averaged over 10 trials. For the clustering model (Section 2.1), we choose $p = 0.6$, $q = 0.4$, and consider two cases of $k = 2$ and 3 cluster problems. Figure 1 (top row) shows that in this model, the number of errors eventually decreases for all $m$, even $m = 2$. This observation is similar to the one in [6]. However, the decrease is much faster for $m = 3$, where accurate partitioning is often observed for $n \geq 100$. We also observe that error rises for larger $k$, thus validating the dependence of the bound on $k$. A similar inference can be drawn from Figure 1 (second row) for the matching problem (Section 2.2), where we use $p = 0.9$, $q = 0.1$ and the number of correct matches as $\sqrt{n}$.

### 5.2 Motion Segmentation on Hopkins 155 dataset

We now turn to practical applications, and test the performance of Algorithm 1 in motion segmentation. We perform the experiments on the Hopkins 155 dataset [24], which contains 120 videos with 2 independent affine motions. Figure 1 (third row) shows two cases, where Algorithm 1 correctly clusters the trajectories into their true groups. We used $4^{th}$-order tensors in the approach, where the large dimensionality of $\widehat{A}_n$ is tackled by using only 500 uniformly sampled columns of $\widehat{A}_n$ for computing $W_n$. We also compare the performance of Algorithm 1, averaged over 20 runs, with some standard approaches. The results for other methods have been taken from [20]. We observe that Algorithm 1 performs reasonably well, while the best performance is obtained using Sparse Grassmann Clustering (SGC) [20], which is expected as SGC is an iterative improvement of Algorithm 1.

### 5.3 Matching point sets from the Mpeg-7 shape database

We now consider a matching problem using points sampled from images in Mpeg-7 database [25]. This problem has been considered in [10]. We use 70 random images, one from each shape class. Ten points were sampled from the boundary of each shape, which formed one point set. The other set of points was generated by adding Gaussian noise of variance $\sigma^2$ to the original points and then using a random affine transformation on the points. In Figure 1 (last row), we compare performance of Algorithm 1 with the methods in [9, 10], which have been shown to outperform other methods. We use 4-way similarities based on ratio of areas of two triangles. We show the variation in the number of correctly detected matches and the F1-score for all methods as $\sigma$ increases from 0 to 0.2. The results show that Algorithm 1 is quite robust compared to [10] in detecting true matches. However, Algorithm 1 does not use additional post-processing as in [9], and hence, allows high number of false positives that reduces F1-score, whereas [9, 10] show similar trends in both plots.

## 6 Concluding remarks

In this paper, we presented a planted partition model for unweighted undirected uniform hypergraphs. We devised a spectral approach (Algorithm 1) for detecting the partitions from the affinity

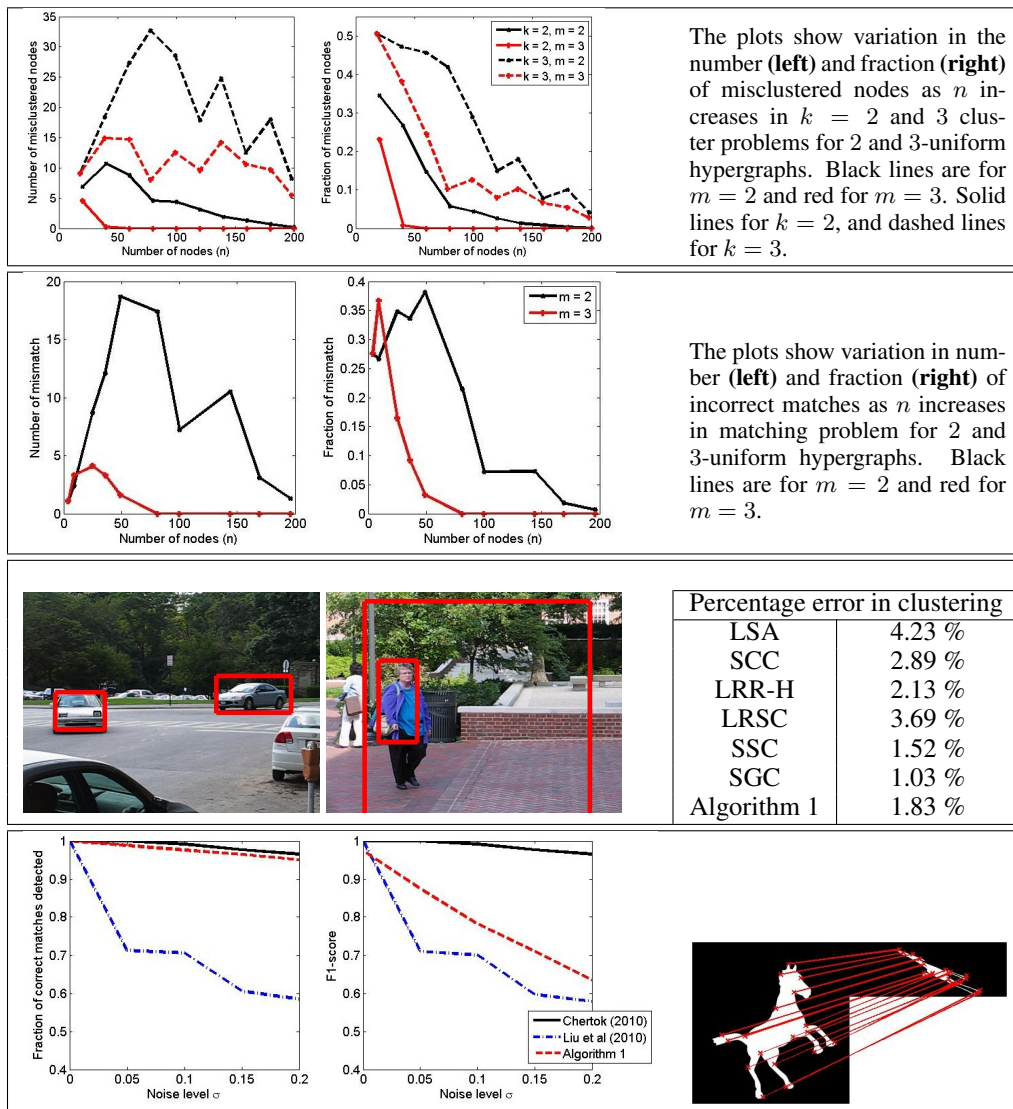

Figure 1: **First row:** Number of misclustered nodes in clustering problem as $n$ increases.
**Second row:** Number of misclustered nodes in matching problem as $n$ increases.
**Third row:** Grouping two affine motions with Algorithm 1 **(left)**, and performance comparison of Algorithm 1 with other methods **(right)**.
**Fourth row:** Variation in number of correct matches detected **(left)** and F1-score **(middle)** as noise level, $\sigma$ increases. **(right)** A pair of images where Algorithm 1 correctly matches all sampled points.

tensor of the corresponding random hypergraph. The above model is appropriate for a number of problems in computer vision including motion segmentation, illumination-invariant face clustering, point-set matching, feature tracking etc. We analyzed the approach to provide an almost sure upper bound on the number of misclustered nodes (c.f. Theorem 1). Using this bound, we conclude that for the problems of subspace clustering and point-set matching, Algorithm 1 is consistent for $m \geq 3$ and $m \geq 4$, respectively. To the best of our knowledge, this is the first theoretical study of the above problems in a probabilistic setting, and also the first theoretical evidence that shows importance of $m$-way affinities.

## Acknowledgement

D. Ghoshdastidar is supported by Google Ph.D. Fellowship in Statistical Learning Theory.

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
