[Supplementary Material]

# Supplementary material

Before, we present the proofs of the results stated in this paper, it will be useful to specify standard quantities and operations used in this paper.

| Notation | Description |
|---|---|
| $I_a$ | $a \times a$ identity matrix |
| $\mathbf{1}_a$ | $a$-dimensional vector of all ones |
| $\mathbf{0}_{a \times b}$ | $a \times b$ zero matrix |
| $\mathsf{P}(\cdot)$ | Probability of an event |
| $\mathsf{E}[\cdot]$ | Expectation of a random quantity |
| $S^{\otimes a}$ | $a$-times Kronecker product of any matrix $S$ with itself |
| $\lambda(S)$ | The set of all eigenvalues of a symmetric matrix $S$ |
| $\lambda_i(S)$ | $i^{th}$ largest eigenvalue of a symmetric matrix $S$ |
| $\widehat{A}$ | $n \times n^{m-1}$ flattened matrix of any $n \times n \times \ldots \times n$ tensor $A$ |

**Proof of Lemma 4**

We begin with computing the expected quantities. Observe that

$$(\mathcal{W}_n)_{ij} = \mathsf{E}[W_n] = \sum_{i_2,\ldots,i_m=1}^{n} \mathsf{E}[(A_n)_{ii_2\ldots i_m}(A_n)_{ji_2\ldots i_m}].$$

For $i \neq j$, the two terms in the expectation are independent, and hence

$$(\mathcal{W}_n)_{ij} = \sum_{i_2,\ldots,i_m=1}^{n} (\mathcal{A}_n)_{ii_2\ldots i_m}(\mathcal{A}_n)_{ji_2\ldots i_m} = (\widehat{\mathcal{A}_n}\widehat{\mathcal{A}_n}^T)_{ij},$$

while for $i = j$, $(A_n)_{ii_2\ldots i_m}^2 = (A_n)_{ii_2\ldots i_m}$, and $(\mathcal{W}_n)_{ii} = \sum_{i_2,\ldots,i_m=1}^{n} (\mathcal{A}_n)_{ii_2\ldots i_m}$. So $\mathcal{W}_n$ and $\widehat{\mathcal{A}_n}\widehat{\mathcal{A}_n}^T$ differ only at the diagonal entries, which implies $\mathcal{P}_n = \mathcal{W}_n - \widehat{\mathcal{A}_n}\widehat{\mathcal{A}_n}^T$ is diagonal with

$$(\mathcal{P}_n)_{ii} = \sum_{i_2,\ldots,i_m=1}^{n} (\mathcal{A}_n)_{ii_2\ldots i_m} - (\mathcal{A}_n)_{ii_2\ldots i_m}^2 = \sum_{i_2,\ldots,i_m=1}^{n} B_{\psi_i\psi_{i_2}\ldots\psi_{i_m}} - B_{\psi_i\psi_{i_2}\ldots\psi_{i_m}}^2$$

using (1). In above summation, we can group nodes in same partition as the terms in the summation are same for these. Define $S_n = Z_n^T Z_n$, whose diagonal entries represent cluster sizes. Then

$$(\mathcal{P}_n)_{ii} = \sum_{j_2,\ldots,j_m=1}^{k} (S_n)_{j_2j_2}\ldots(S_n)_{j_mj_m}(B_{\psi_i j_2\ldots j_m} - B_{\psi_i j_2\ldots j_m}^2)$$

$$= \sum_{l=1}^{k^{m-1}} (S_n^{\otimes(m-1)})_{ll}\left(\widehat{B}_{\psi_i l} - \widehat{B}_{\psi_i l}^2\right). \tag{8}$$

Thus, we see that all diagonal entries of $\mathcal{P}_n$ are not distinct, rather, there are only $k$ distinct terms and $(\mathcal{P}_n)_{ii} = (\mathcal{P}_n)_{jj}$, whenever $\psi_i = \psi_j$. Similarly, we compute $(\mathcal{D}_n)_{ii}$ as

$$(\mathcal{D}_n)_{ii} = \sum_{l=1}^{k^{m-1}} (S_n^{\otimes(m-1)})_{ll}\widehat{B}_{\psi_i l}\left(1 + ((S_n)_{\psi_i\psi_i} - 1)\widehat{B}_{\psi_i l} + \sum_{j\neq\psi_i}(S_n)_{jj}\widehat{B}_{jl}\right), \tag{9}$$

which reveals that $(\mathcal{D}_n)_{ii} = (\mathcal{D}_n)_{jj}$, whenever $\psi_i = \psi_j$. Hence, we can define $\widetilde{\mathcal{D}}_n, \widetilde{\mathcal{P}}_n \in \mathbb{R}^{k \times k}$ such that $(\mathcal{D}_n)_{ii} = (\widetilde{\mathcal{D}}_n)_{\psi_i\psi_i}$ and $(\mathcal{P}_n)_{ii} = (\widetilde{\mathcal{P}}_n)_{\psi_i\psi_i}$, and it is easy to see that $\mathcal{D}_n Z_n = Z_n \widetilde{\mathcal{D}}_n$ and $\mathcal{P}_n Z_n = Z_n \widetilde{\mathcal{P}}_n$. With these terms, we characterize the eigenvalues and eigenvectors of $\mathcal{L}_n$.

Let $U \in \mathbb{R}^{k \times k}$ be orthonormal eigenvector matrix of the matrix $\widetilde{\mathcal{D}}_n^{-1/2}(C_n + \widetilde{\mathcal{P}}_n)\widetilde{\mathcal{D}}_n^{-1/2}$, i.e.,

$$\widetilde{\mathcal{D}}_n^{-1/2}(C_n + \widetilde{\mathcal{P}}_n)\widetilde{\mathcal{D}}_n^{-1/2}U = U\Lambda_1,$$

with $\Lambda_1$ being diagonal with entries being eigenvalues of the stated matrix. Then, for $\mu_n = S_n^{-1/2}U$,

$$
\begin{aligned}
\mathcal{L}_n Z_n \mu_n &= \mathcal{D}_n^{-1/2}(\widehat{\mathcal{A}_n}\widehat{\mathcal{A}_n}^T + \mathcal{P}_n)\mathcal{D}_n^{-1/2}Z_n\mu_n \\
&= \mathcal{D}_n^{-1/2}(Z_n\widehat{B}S_n^{\otimes(m-1)}\widehat{B}^T Z_n^T + \mathcal{P}_n)\mathcal{D}_n^{-1/2}Z_n S_n^{-1/2}U \\
&= \mathcal{D}_n^{-1/2}Z_n\widehat{B}S_n^{\otimes(m-1)}\widehat{B}^T S_n^{1/2}\widetilde{\mathcal{D}}_n^{-1/2}U + Z_n S_n^{-1/2}\widetilde{\mathcal{D}}_n^{-1/2}\widetilde{\mathcal{P}}_n\widetilde{\mathcal{D}}_n^{-1/2}U \\
&= Z_n S_n^{-1/2}\widetilde{\mathcal{D}}_n^{-1/2}(C_n + \widetilde{\mathcal{P}}_n)\widetilde{\mathcal{D}}_n^{-1/2}U \\
&= Z_n S_n^{-1/2}U\Lambda_1 = Z_n\mu_n\Lambda_1 \ .
\end{aligned}
$$

Thus, columns of $Z_n\mu_n$ are $k$ eigenvectors of $\mathcal{L}_n$, with corresponding eigenvalues being the diagonal entries in $\Lambda_1$. It is also easy to see that columns of $Z_n\mu_n$ are orthonormal. We note here that for computing $\widehat{\mathcal{A}_n}\widehat{\mathcal{A}_n}^T$, we have used some properties of Kronecker products such as $(X^T)^{\otimes l} = (X^{\otimes l})^T$ and $(X^{\otimes l})(X_1^{\otimes l}) = (XX_1)^{\otimes l}$.

We now focus on the remaining $(n-k)$ eigenvalues of $\mathcal{L}_n$. From its definition, one can see $\mathcal{L}_n$ is symmetric positive semi-definite. Hence, its $n$ eigenvectors form an orthonormal basis. Let $Y \in \mathbb{R}^{n\times(n-k)}$ be such that columns of $Y$ are remaining orthonormal eigenvectors, *i.e.*, $\mathcal{L}_n Y = Y\Lambda_2$ for some diagonal matrix of eigenvalues $\Lambda_2 \in \mathbb{R}^{(n-k)\times(n-k)}$. Now, since $\mu_n = S_n^{-1/2}U$ has rank-$k$, hence $k$ columns of $Z_n\mu_n$ span the range space of $Z_n$. So columns of $Y$ must span the null space of $Z_n^T$. Thus, $Z_n^T Y = 0$ and

$$
Y\Lambda_2 = \mathcal{L}_n Y = \mathcal{D}_n^{-1/2}(Z_n\widehat{B}S_n^{\otimes(m-1)}\widehat{B}^T Z_n^T + \mathcal{P}_n)\mathcal{D}_n^{-1/2}Y = \mathcal{D}_n^{-1}\mathcal{P}_n Y.
$$

Thus, columns of $Y$ are eigenvectors of $\mathcal{D}_n^{-1}\mathcal{P}_n$ and the eigenvalues in $\Lambda_2$ are a subset of the diagonal entries of $\mathcal{D}_n^{-1}\mathcal{P}_n$, which are $\frac{(\mathcal{P}_n)_{ii}}{(\mathcal{D}_n)_{ii}}$, $i = 1,\ldots,n$, which also same as $\frac{(\widetilde{\mathcal{P}}_n)_{ii}}{(\widetilde{\mathcal{D}}_n)_{ii}}$, $i = 1,\ldots,k$.

Though we know that columns of $Z_n\mu_n$ are orthonormal eigenvectors of $\mathcal{L}_n$, we still need to ensure that they are the leading eigenvectors, *i.e.*, the eigenvalues in $\Lambda_1$ are strictly greater than the ones in $\Lambda_2$. It is sufficient to satisfy

$$
\lambda_k\big(\widetilde{\mathcal{D}}_n^{-1/2}(C_n + \widetilde{\mathcal{P}}_n)\widetilde{\mathcal{D}}_n^{-1/2}\big) > \max_{1\leq i\leq k}\frac{(\widetilde{\mathcal{P}}_n)_{ii}}{(\widetilde{\mathcal{D}}_n)_{ii}},
$$

where $\lambda_k(\cdot)$ denotes $k^{th}$ largest eigenvalue, which is in fact smallest eigenvalue in this case. Now we can consider the matrix on the left as the matrix $\widetilde{\mathcal{D}}_n^{-1/2}C_n\widetilde{\mathcal{D}}_n^{-1/2}$ perturbed by the matrix $\widetilde{\mathcal{D}}_n^{-1}\widetilde{\mathcal{P}}_n$. Then by Weyl's inequality [21], we can say

$$
\max_{1\leq i\leq k}\left|\lambda_i\big(\widetilde{\mathcal{D}}_n^{-1/2}(C_n + \widetilde{\mathcal{P}}_n)\widetilde{\mathcal{D}}_n^{-1/2}\big) - \lambda_i\big(\widetilde{\mathcal{D}}_n^{-1/2}C_n\widetilde{\mathcal{D}}_n^{-1/2}\big)\right| \leq \lambda_1\big(\widetilde{\mathcal{D}}_n^{-1}\widetilde{\mathcal{P}}_n\big)
$$

and hence,

$$
\lambda_k\big(\widetilde{\mathcal{D}}_n^{-1/2}(C_n + \widetilde{\mathcal{P}}_n)\widetilde{\mathcal{D}}_n^{-1/2}\big) \geq \lambda_k\big(\widetilde{\mathcal{D}}_n^{-1/2}C_n\widetilde{\mathcal{D}}_n^{-1/2}\big) - \max_{1\leq i\leq k}\frac{(\widetilde{\mathcal{P}}_n)_{ii}}{(\widetilde{\mathcal{D}}_n)_{ii}}.
$$

Thus it is enough to satisfy $\lambda_k\big(\widetilde{\mathcal{D}}_n^{-1/2}C_n\widetilde{\mathcal{D}}_n^{-1/2}\big) > 2\max_i \frac{(\widetilde{\mathcal{P}}_n)_{ii}}{(\widetilde{\mathcal{D}}_n)_{ii}}$ to ensure that $Z_n\mu_n$ contain leading eigenvectors of $\mathcal{L}_n$. Now,

$$
\max_{1\leq i\leq k}\frac{(\widetilde{\mathcal{P}}_n)_{ii}}{(\widetilde{\mathcal{D}}_n)_{ii}} \leq \frac{\max\limits_{1\leq i\leq n}(\widetilde{\mathcal{P}}_n)_{ii}}{\underline{\mathcal{D}}_n} \leq \frac{1}{\underline{\mathcal{D}}_n}\sum_{j_2,\ldots,j_m=1}^{k}(S_n)_{j_2 j_2}\ldots(S_n)_{j_m j_m} = \frac{n^{m-1}}{\underline{\mathcal{D}}_n},
$$

where we use (8) along with the fact that $(\widehat{B}_{il} - \widehat{B}_{il}^2) \in [0,1]$ for all $i,l$. On the other hand,

$$
\lambda_k\big(\widetilde{\mathcal{D}}_n^{-1/2}C_n\widetilde{\mathcal{D}}_n^{-1/2}\big) = \min_{y\in\mathbb{R}^k}\frac{y^T C_n y}{y^T\widetilde{\mathcal{D}}_n y} \geq \frac{1}{\overline{\mathcal{D}}_n}\min_{y\in\mathbb{R}^k}\frac{y^T C_n y}{y^T y} = \frac{\lambda_k(C_n)}{\overline{\mathcal{D}}_n}.
$$

Thus, $\delta_n = \left(\frac{\lambda_k(C_n)}{\overline{\mathcal{D}}_n} - \frac{2n^{m-1}}{\underline{\mathcal{D}}_n}\right) > 0$ is a sufficient condition for columns of $Z_n\mu_n$ to be leading eigenvectors of $\mathcal{L}_n$. The last part in Lemma 4 follows from the structure of $Z_n$. Since, $\mu_n$ is rank-$k$, it has $k$ distinct rows, and one can see that $Z_n\mu_n$ contains the $k$ distinct rows of $\mu_n$, each unique row being a representative of a partition.

**Proof of Lemma 5**

For a particular $n$, define $\beta = \frac{\log n}{n^{\frac{m-1}{2}}}$, $\Gamma_1 = \bigcap_{i=1}^{n} \left\{ (D_n)_{ii} \in (\mathcal{D}_n)_{ii}[1-\beta, 1+\beta] \right\}$ and

$\Gamma_2 = \bigcap_{i,j=1}^{n} \left\{ |(W_n)_{ij} - (\mathcal{W}_n)_{ij}| \leq n^{\frac{m-1}{2}} \log n \right\}$. Now,

$$
\mathsf{P}\left( \|L_n - \mathcal{L}_n\|_F > \frac{4n^{\frac{m+1}{2}} \log n}{\underline{\mathcal{D}}_n} \right)
$$

$$
\leq \mathsf{P}\left( \left\{ \|L_n - \mathcal{L}_n\|_F^2 > \left( \frac{4n^{\frac{m+1}{2}} \log n}{\underline{\mathcal{D}}_n} \right)^2 \right\} \bigcap \Gamma_1 \bigcap \Gamma_2 \right) + \mathsf{P}\left( (\Gamma_1 \bigcap \Gamma_2)^c \right)
$$

$$
\leq \mathsf{P}\left( \left\{ \sum_{i,j=1}^{n} |(L_n)_{ij} - (\mathcal{L}_n)_{ij}|^2 > \left( \frac{4n^{\frac{m+1}{2}} \log n}{\underline{\mathcal{D}}_n} \right)^2 \right\} \bigcap \Gamma_1 \bigcap \Gamma_2 \right) + \mathsf{P}\left( \Gamma_1^c \right) + \mathsf{P}\left( \Gamma_2^c \right)
$$

$$
\leq \mathsf{P}\left( \bigcup_{i,j=1}^{n} \left\{ |(L_n)_{ij} - (\mathcal{L}_n)_{ij}| > \frac{1}{n} \frac{4n^{\frac{m+1}{2}} \log n}{\underline{\mathcal{D}}_n} \right\} \bigcap \Gamma_1 \bigcap \Gamma_2 \right) + \mathsf{P}\left( \Gamma_1^c \right) + \mathsf{P}\left( \Gamma_2^c \right)
$$

$$
\leq \sum_{i,j=1}^{n} \mathsf{P}\left( \left\{ |(L_n)_{ij} - (\mathcal{L}_n)_{ij}| > \frac{4n^{\frac{m-1}{2}} \log n}{\underline{\mathcal{D}}_n} \right\} \bigcap \Gamma_1 \bigcap \Gamma_2 \right) + \mathsf{P}\left( \Gamma_1^c \right) + \mathsf{P}\left( \Gamma_2^c \right)
$$

Defining $\widetilde{L}_n = \mathcal{D}_n^{-1/2} W_n \mathcal{D}_n^{-1/2}$, we have $|(L_n)_{ij} - (\mathcal{L}_n)_{ij}| \leq |(L_n)_{ij} - (\widetilde{L}_n)_{ij}| + |(\widetilde{L}_n)_{ij} - (\mathcal{L}_n)_{ij}|$, and hence,

$$
\mathsf{P}\left( \|L_n - \mathcal{L}_n\|_F > \frac{4n^{\frac{m+1}{2}} \log n}{\underline{\mathcal{D}}_n} \right) \leq \sum_{i,j=1}^{n} \mathsf{P}\left( \left\{ |(L_n)_{ij} - (\widetilde{L}_n)_{ij}| > \frac{2n^{\frac{m-1}{2}} \log n}{\underline{\mathcal{D}}_n} \right\} \bigcap \Gamma_1 \right)
$$

$$
+ \sum_{i,j=1}^{n} \mathsf{P}\left( \left\{ |(\widetilde{L}_n)_{ij} - (\mathcal{L}_n)_{ij}| > \frac{2n^{\frac{m-1}{2}} \log n}{\underline{\mathcal{D}}_n} \right\} \bigcap \Gamma_2 \right) + \mathsf{P}\left( \Gamma_1^c \right) + \mathsf{P}\left( \Gamma_2^c \right). \quad (10)
$$

Dealing with the second summation in (10) is easy as, on $\Gamma_2$,

$$
|(\widetilde{L}_n)_{ij} - (\mathcal{L}_n)_{ij}| = \frac{1}{\sqrt{(\mathcal{D}_n)_{ii}(\mathcal{D}_n)_{jj}}} |(W_n)_{ij} - (\mathcal{W}_n)_{ij}|
$$

$$
\leq \frac{n^{\frac{m-1}{2}} \log n}{\sqrt{(\mathcal{D}_n)_{ii}(\mathcal{D}_n)_{jj}}} \leq \frac{n^{\frac{m-1}{2}} \log n}{\underline{\mathcal{D}}_n}.
$$

Thus, the second term in (10) is zero. Now, if $\Gamma_3 = \bigcap_{i,j=1}^{n} \left\{ \frac{1}{\sqrt{(D_n)_{ii}(D_n)_{jj}}} \in \frac{[1-2\beta, 1+2\beta]}{\sqrt{(\mathcal{D}_n)_{ii}(\mathcal{D}_n)_{jj}}} \right\}$,

then we claim that $\Gamma_1 \subset \Gamma_3$ for $\beta \leq \frac{1}{2}$. This is obvious because for any case in $\Gamma_1$, we can write

$$
\frac{1}{\sqrt{(D_n)_{ii}(D_n)_{jj}}} \in \frac{1}{\sqrt{(\mathcal{D}_n)_{ii}(\mathcal{D}_n)_{jj}}} \left[ \frac{1}{1+\beta}, \frac{1}{1-\beta} \right],
$$

where

$$
\frac{1}{1+\beta} = 1 - \frac{\beta}{1+\beta} \geq 1 - \beta \geq 1 - 2\beta, \qquad \text{and} \qquad \frac{1}{1-\beta} = 1 + \frac{\beta}{1-\beta} \leq 1 + 2\beta
$$

for $\beta \leq \frac{1}{2}$. Thus, the claim. Note that the condistion $\beta \leq \frac{1}{2}$ is not restrictive as it always holds for $m > 2$, and eventually holds for $m = 2$. Therefore, on $\Gamma_1 \subset \Gamma_3$

$$
|(L_n)_{ij} - (\widetilde{L}_n)_{ij}| = |(W_n)_{ij}| \left| \frac{1}{\sqrt{(D_n)_{ii}(D_n)_{jj}}} - \frac{1}{\sqrt{(\mathcal{D}_n)_{ii}(\mathcal{D}_n)_{jj}}} \right|
$$

$$
\leq n^{m-1} \frac{2\beta}{\sqrt{(\mathcal{D}_n)_{ii}(\mathcal{D}_n)_{jj}}} \leq \frac{2n^{\frac{m-1}{2}} \log n}{\underline{\mathcal{D}}_n},
$$

using the fact $|(W_n)_{ij}| = (\widehat{A}_n \widehat{A}_n^T)_{ij} \le n^{m-1}$ as the quantity is a summation of $n^{m-1}$ binary terms. Thus, we see that the first summation in (10) is also zero.

To compute an upper bound for $\mathsf{P}(\Gamma_1^c)$, we observe

$$(D_n)_{ii} - (\mathcal{D}_n)_{ii} = \sum_{i_2,\ldots,i_m=1}^{n} \left( (A_n)_{ii_2\ldots i_m} - (\mathcal{A}_n)_{ii_2\ldots i_m} \right)$$
$$+ \sum_{j\neq i} \sum_{i_2,\ldots,i_m=1}^{n} \left( (A_n)_{ii_2\ldots i_m}(A_n)_{ji_2\ldots i_m} - (\mathcal{A}_n)_{ii_2\ldots i_m}(\mathcal{A}_n)_{ji_2\ldots i_m} \right).$$

In each summation, all the terms are not independent since for any $i_2\ldots i_m$, all its permutations give the same random variable. However, we can avoid such repetitions by summing only over $i_2 \le \ldots \le i_m$, where for all $i_2\ldots i_m$, there can be at most $(m-1)!$ of the same quantity. Hence,

$$|(D_n)_{ii} - (\mathcal{D}_n)_{ii}| \le (m-1)! \left| \sum_{i_2\le\ldots\le i_m} \left( (A_n)_{ii_2\ldots i_m} - (\mathcal{A}_n)_{ii_2\ldots i_m} \right) \right|$$
$$+ (m-1)! \sum_{j\neq i} \left| \sum_{i_2\le\ldots\le i_m} \left( (A_n)_{ii_2\ldots i_m}(A_n)_{ji_2\ldots i_m} - (\mathcal{A}_n)_{ii_2\ldots i_m}(\mathcal{A}_n)_{ji_2\ldots i_m} \right) \right|$$

where $\{(A_n)_{ii_2\ldots i_m}\}_{i_2\le\ldots\le i_m}$ and for each $j\neq i$ $\{(A_n)_{ii_2\ldots i_m}(A_n)_{ji_2\ldots i_m}\}_{i_2\le\ldots\le i_m}$ form independent sequences of $\binom{n+m-2}{m-1}$ random variables, where each random variable lies in $[0,1]$. Further, the quantities involving $\mathcal{A}_n$ are exactly expected values of the terms beside them. Thus,

$$\mathsf{P}\big(|(D_n)_{ii} - (\mathcal{D}_n)_{ii}| > \beta(\mathcal{D}_n)_{ii}\big)$$
$$\le \mathsf{P}\left( \sum_{j=1}^{n} \left| \sum_{i_2\le\ldots\le i_m} (A_n)_{ii_2\ldots i_m}(A_n)_{ji_2\ldots i_m} - \mathsf{E}[(A_n)_{ii_2\ldots i_m}(A_n)_{ji_2\ldots i_m}] \right| > \frac{(\mathcal{D}_n)_{ii}\beta}{(m-1)!} \right)$$
$$\le \sum_{j=1}^{n} \mathsf{P}\left( \left| \sum_{i_2\le\ldots\le i_m} (A_n)_{ii_2\ldots i_m}(A_n)_{ji_2\ldots i_m} - \mathsf{E}[(A_n)_{ii_2\ldots i_m}(A_n)_{ji_2\ldots i_m}] \right| > \frac{(\mathcal{D}_n)_{ii}\log n}{(m-1)!n^{\frac{m+1}{2}}} \right).$$

We can upper bound each of the above probabilities using Hoeffding's inequality to get

$$\mathsf{P}\big(|(D_n)_{ii} - (\mathcal{D}_n)_{ii}| > \beta(\mathcal{D}_n)_{ii}\big) \le 2n \exp\left( -\frac{2}{\binom{n+m-2}{m-1}} \frac{(\mathcal{D}_n)_{ii}^2(\log n)^2}{((m-1)!)^2 n^{m+1}} \right)$$
$$\le 2n^{1-2(\log n)\left(\frac{\underline{\mathcal{D}}_n}{(m-1)!n^m}\right)^2}$$

since $\underline{\mathcal{D}}_n \le (\mathcal{D}_n)_{ii}$ and $\binom{n+m-2}{m-1} \le n^{m-1}$. With this result, we have

$$\mathsf{P}(\Gamma_1^c) \le \sum_{i=1}^{n} \mathsf{P}\big(|(D_n)_{ii} - (\mathcal{D}_n)_{ii}| > \beta(\mathcal{D}_n)_{ii}\big) \le 2n^{2-2(\log n)\left(\frac{\underline{\mathcal{D}}_n}{(m-1)!n^m}\right)^2}.$$

We proceed in a similar way to bound $\mathsf{P}(\Gamma_2^c)$ as

$$\mathsf{P}(\Gamma_2^c) \le \sum_{i,j=1}^{n} \mathsf{P}\big(|(W_n)_{ij} - (\mathcal{W}_n)_{ii}| > n^{\frac{m-1}{2}}\log n\big)$$
$$= \sum_{i,j=1}^{n} \mathsf{P}\left( \left| \sum_{i_2,\ldots,i_m=1}^{n} (A_n)_{ii_2\ldots i_m}(A_n)_{ji_2\ldots i_m} - \mathsf{E}[(A_n)_{ii_2\ldots i_m}(A_n)_{ji_2\ldots i_m}] \right| > n^{\frac{m-1}{2}}\log n \right)$$
$$\le \sum_{i,j=1}^{n} \mathsf{P}\left( \left| \sum_{i_2\le\ldots\le i_m} (A_n)_{ii_2\ldots i_m}(A_n)_{ji_2\ldots i_m} - \mathsf{E}[(A_n)_{ii_2\ldots i_m}(A_n)_{ji_2\ldots i_m}] \right| > \frac{n^{\frac{m-1}{2}}\log n}{(m-1)!} \right).$$

Now, by Hoeffding's inequality,

$$\mathsf{P}(\Gamma_2^c) \leq 2n^2 \exp\left(-\frac{2}{\binom{n+m-2}{m-1}} \frac{n^{m-1}(\log n)^2}{((m-1)!)^2}\right)$$

$$\leq 2n^{2-2\frac{\log n}{((m-1)!)^2}} \leq 2n^{2-2(\log n)\left(\frac{\mathcal{D}_n}{(m-1)!n^m}\right)^2}.$$

since $\underline{\mathcal{D}}_n \leq n^m$. Putting these bounds in (10), we have for any $n$

$$\mathsf{P}\left(\|L_n - \mathcal{L}_n\|_F > \frac{4n^{\frac{m+1}{2}}\log n}{\underline{\mathcal{D}}_n}\right) \leq 4n^{2-2(\log n)\left(\frac{\mathcal{D}_n}{(m-1)!n^m}\right)^2}.$$

if $\beta \leq \frac{1}{2}$. Now, if there exists $N$ such that $\left(\frac{\mathcal{D}_n}{(m-1)!n^m}\right)^2 \geq \frac{2}{\log n}$ for all $n > N$, then

$$\sum_{n=1}^{\infty}\mathsf{P}\left(\|L_n - \mathcal{L}_n\|_F > \frac{4n^{\frac{m+1}{2}}\log n}{\underline{\mathcal{D}}_n}\right) \leq N + 4\sum_{n=N+1}^{\infty} n^{2-4} < \infty,$$

and hence, by Borel-Cantelli lemma,

$$\mathsf{P}\left(\left\{\|L_n - \mathcal{L}_n\|_F > \frac{4n^{\frac{m+1}{2}}\log n}{\underline{\mathcal{D}}_n}\right\} \text{ i.o.}\right) = 0,$$

which implies $\|L_n - \mathcal{L}_n\|_F \leq \frac{4n^{\frac{m+1}{2}}\log n}{\underline{\mathcal{D}}_n}$ a.s.

**Proof of Lemma 6**

We begin by observing that $\delta_n$ defined in Lemma 4 is a lower bound on the eigen-gap $\lambda_k(\mathcal{L}_n) - \lambda_{k+1}(\mathcal{L}_n)$. By Weyl's inequality,

$$\max_{1 \leq i \leq n}|\lambda_i(L_n) - \lambda_i(\mathcal{L}_n)| \leq \|L_n - \mathcal{L}_n\|_2 \leq \|L_n - \mathcal{L}_n\|_F = O\left(\frac{(\log n)^{3/2}}{n^{\frac{m-1}{2}}}\right) \text{ a.s.}$$

under the condition on $\underline{\mathcal{D}}_n$ stated in Lemma 5. Thus if $\frac{(\log n)^{3/2}}{n^{\frac{m-1}{2}}} = o(\delta_n)$ (condition in Lemma 6), then

$$\max_{1 \leq i \leq n}|\lambda_i(L_n) - \lambda_i(\mathcal{L}_n)| = o(\delta_n) = o(\lambda_k(\mathcal{L}_n) - \lambda_{k+1}(\mathcal{L}_n)) \text{ a.s.}$$

Define an interval $\mathcal{I}_n = \left[\frac{1}{2}(\lambda_k(\mathcal{L}_n) + \lambda_{k+1}(\mathcal{L}_n)), 1\right]$, then

$$\min\{|\lambda(\mathcal{L}_n) \cap \mathcal{I}_n - \mathcal{I}_n^c|\} = \min\{|\lambda(\mathcal{L}_n) \cap \mathcal{I}_n^c - \mathcal{I}_n|\} \geq \frac{\delta_n}{2},$$

where the above minimum is the minimum absolute difference between any two elements taken from either sets. From above, we see that this quantity decays much slowly than the difference between the eigenvalues of $L_n$ and $\mathcal{L}_n$. Thus, eventually we can argue

$$|\{\lambda(L_n) \cap \mathcal{I}_n\}| = |\{\lambda(\mathcal{L}_n) \cap \mathcal{I}_n\}| = k \text{ a.s.}$$

Hence, by the modified version of Davis-Kahan theorem [6, Theorem 2.1], we have that there exists orthonormal matrix $O_n \in \mathbb{R}^{k \times k}$ such that

$$\|X_n - Z_n\mu_nO_n\|_F \leq \frac{2\|L_n - \mathcal{L}_n\|_F}{\frac{1}{2}(\lambda_k(\mathcal{L}_n) - \lambda_{k+1}(\mathcal{L}_n))}$$

$$\leq \frac{4\|L_n - \mathcal{L}_n\|_F}{\delta_n} \leq \frac{16n^{\frac{m+1}{2}}\log n}{\delta_n\underline{\mathcal{D}}_n} \text{ a.s.}$$

Here, $O_n$ can be computed as follows. Let $\mu_n^T Z_n^T X_n \in \mathbb{R}^{k \times k}$ has the singular value decomposition as $\mu_n^T Z_n^T X_n = U\Sigma V^T$, then one can show [6] that $O_n = UV^T$.

**Proof of Lemma 8**

This proof is stated for completeness, but it can be found as part of [6, Theorem 3.1]. Let $C \in \mathbb{R}^{n \times k}$ contain the $k$ centroids in its rows, *i.e.*, $C^T = [c_1^T \ldots c_n^T]$. Observe that one can write $C$ from the objective of $k$-means algorithm as

$$C = \operatorname*{argmin}_{C'} \|X_n - C'\|_F^2,$$

where the minimum is taken over all $n \times k$ matrices with no more than $k$ distinct rows. Since, $Z_n \mu_n O_n$ also has $k$ distinct rows, hence $\|X_n - C\|_F \leq \|X_n - Z_n \mu_n O_n\|_F$. Thus,

$$\|C - Z_n \mu_n O_n\|_F \leq \|X_n - Z_n \mu_n O_n\|_F + \|X_n - C\|_F \leq 2\|X_n - Z_n \mu_n O_n\|_F$$

One the other hand, we can write the number of misclustered nodes as

$$|M_n| = \sum_{i \in M_n} 1 \leq 2\overline{S}_n \sum_{i \in M_n} \|c_i - z_i \mu_n O_n\|_2^2$$

from the definition of $M_n$. An upper bound can be found by summing over all $i = 1, \ldots, n$. So,

$$|M_n| \leq 2\overline{S}_n \|C - Z_n \mu_n O_n\|_F^2 \leq 8\overline{S}_n \|X_n - Z_n \mu_n O_n\|_F^2.$$

**Proof of Corollary 2**

We begin by computing the eigenvalues of $C_n$. Observe that $S_n = \frac{n}{k} I_k$, and hence

$$C_n = S_n^{1/2} \widehat{B} S_n^{\otimes(m-1)} \widehat{B}^T S_n^{1/2} = \frac{n^m}{k^m} \widehat{B}\widehat{B}^T,$$

where $\widehat{B}$ can be written, after rearranging the columns, as

$$\widehat{B} = q \mathbf{1}_k \mathbf{1}_{k^{m-1}} + (p - q)[I_k \ \mathbf{0}_{k \times k^{m-1}}].$$

Thus, we can write

$$C_n = \frac{n^m}{k^m} \left(k^{m-1} q^2 \mathbf{1}_k \mathbf{1}_k^T + 2q(p-q) \mathbf{1}_k \mathbf{1}_k^T + (p-q)^2 I_k\right).$$

From above one can easily see that the leading eigenvector of $C_n$ is $\mathbf{1}_k$ with eigenvalue

$$\lambda_1(C_n) = n^m q^2 + \frac{2n^m q(p-q)}{k^{m-1}} + \frac{n^m (p-q)^2}{k^m},$$

while all the other eigenvalues, whose eigenvectors are orthogonal to $\mathbf{1}_k$, are

$$\lambda_2(C_n) = \ldots = \lambda_k(C_n) = \frac{n^m (p-q)^2}{k^m}.$$

Moreover $\underline{\mathcal{D}}_n = \overline{\mathcal{D}}_n \in [q^2 n^m, n^m]$, which shows that second condition in Theorem 1 is satisfied beyond some $N$. Also,

$$\delta_n = \frac{\lambda_k(C_n) - 2n^{m-1}}{\underline{\mathcal{D}}_n} \geq \frac{\frac{n^m}{k^m}(p-q)^2 - 2n^{m-1}}{n^m} = \frac{1}{k^m}\left((p-q)^2 - \frac{2k^m}{n}\right).$$

Whenever, $k \leq \left(\frac{n(p-q)^2}{4}\right)^{\frac{1}{m}}$, we have $\delta_n \geq \frac{(p-q)^2}{2k^m} > 0$. Thus, if $k = O(n^{\frac{1}{2m}}(\log n)^{-1}) = o(n^{\frac{1}{m}})$, then $\delta_n > 0$ eventually holds. Finally, for all $m \geq 2$,

$$(\log n)^{3/2} = o\left(\frac{(\log n)^m}{n^{1/2}} n^{\frac{m-1}{2}}\right) = o\left(\frac{n^{\frac{m-1}{2}}}{k^m}\right) = o(\delta_n n^{\frac{m-1}{2}})$$

satisfying the last condition in Theorem 1. Hence, we can use the bound on $|M_n|$ to get

$$|M_n| = O\left(\frac{\frac{n}{k}(\log n)^2 n^{m+1}}{\frac{(p-q)^4}{4k^{2m}} q^4 n^{2m}}\right) = O\left(\frac{k^{2m-1}(\log n)^2}{n^{m-2}}\right) = O\left(\frac{(\log n)^{3-2m}}{n^{m-3+\frac{1}{2m}}}\right) \quad \text{a.s.}$$

For $m = 2$, $\frac{|M_n|}{n} = O\left(\frac{n^{-1/4}}{\log n}\right)$, which eventually vanishes a.s., whereas for $m \geq 3$, $|M_n| = o\left(\frac{1}{n^{m-3}(\log n)^m}\right)$ a.s. proving consistency of the method in this case.

**Proof of Corollary 3**

Note here that $C_n, S_n \in \mathbb{R}^{2 \times 2}$ with $(S_n)_{11} = s$ and $(S_n)_{22} = n - s$. In matching problem, $s = \sqrt{n}$, but we will keep the proof general. Now,

$$\lambda_2(C_n) = \min_{y \in \mathbb{R}^k} \frac{y^T C_n y}{y^T y} = \min_{y \in \mathbb{R}^k} \frac{y^T \widehat{B} S_n^{\otimes(m-1)} \widehat{B}^T y}{y^T S_n^{-1} y} \geq s \min_{y \in \mathbb{R}^k} \frac{y^T \widehat{B} S_n^{\otimes(m-1)} \widehat{B}^T y}{y^T y}$$

since we have $s < (n - s)$. Observing than $\widehat{B}_{ij} \geq q$ for all $i, j$, we can write the numerator as

$$y^T \widehat{B} S_n^{\otimes(m-1)} \widehat{B}^T y = \sum_{j=1}^{2^{m-1}} (S_n^{\otimes(m-1)})_{jj} \left( \sum_{i=1}^{2} \widehat{B}_{ij} y_i \right)^2 \geq q^2 (y^T y) \sum_{j=1}^{2^{m-1}} (S_n^{\otimes(m-1)})_{jj},$$

where the summation equals $n^{m-1}$. Thus, $\lambda_2(C_n) \geq q^2 s n^{m-1}$. So to ensure $\delta_n > 0$, we need to satisfy $s > \frac{2}{q^2} \frac{\overline{\mathcal{D}}_n}{\underline{\mathcal{D}}_n}$.

Now, using (9), we can write

$$\overline{\mathcal{D}}_n = (\widetilde{\mathcal{D}}_n)_{11} = s^m p(1 + (s-1)p + (n-s)q) + \gamma = s^m p(1 - p + \eta) + \gamma,$$

$$\text{and } \underline{\mathcal{D}}_n = (\widetilde{\mathcal{D}}_n)_{22} = s^m q(1 + (n-s-1)q + sq) + \gamma = s^m q(1 - q + \eta) + \gamma,$$

$$\text{where } \eta = nq + s(p - q) \quad \text{and} \quad \gamma = \sum_{j=2}^{2^{m-1}} (S_n^{\otimes(m-1)})_{jj} q(1 - q + nq).$$

Observe that both $\gamma, \eta > 0$. Using the relation $\frac{a+b}{c+d} \leq \max\{\frac{a}{c} + \frac{b}{d}\}$, which holds when all quantities are positive, we can write

$$\frac{\overline{\mathcal{D}}_n}{\underline{\mathcal{D}}_n} \leq \max \left\{ \frac{s^m p(1 - p + \eta)}{s^m q(1 - q + \eta)}, \frac{\gamma}{\gamma} \right\} \leq \max \left\{ \frac{p}{q} \max \left\{ \frac{1 - p}{1 - q}, 1 \right\}, 1 \right\}.$$

Since, $p > q$, i.e., $(1 - p) < (1 - q)$, we have $\frac{\overline{\mathcal{D}}_n}{\underline{\mathcal{D}}_n} \leq \frac{p}{q}$. Hence, for $s \geq \frac{3p}{q^3}$,

$$\delta_n = \frac{n^{m-1}}{\overline{\mathcal{D}}_n} \left( q^2 s - 2 \frac{\overline{\mathcal{D}}_n}{\underline{\mathcal{D}}_n} \right) \geq \frac{n^{m-1}}{\overline{\mathcal{D}}_n} \left( q^2 \frac{3p}{q^3} - 2\frac{p}{q} \right) \geq \frac{p}{nq} > 0$$

since $\overline{\mathcal{D}}_n \leq n^m$, thereby satisfying first condition of Theorem 1. Also, it is easy to see that $\underline{\mathcal{D}}_n \geq q^2 n^m$ which implies second condition holds eventually. Further, since $(\log n)^{3/2} = o(n^{\frac{m-3}{2}}) = o(\delta_n n^{\frac{m-1}{2}})$ for all $m > 4$, we can use bound in Theorem 1 to claim

$$|M_n| = O\left( \frac{(\log n)^2 n^{m+2}}{\frac{p^2}{q^2 n^2} n^{2m}} \right) = O\left( \frac{(\log n)^2}{n^{m-4}} \right) \text{ a.s.}$$

showing that the approach is consistent for all $m > 4$ when the size of the strongly connected component is at least $\frac{3p}{q^3}$.

Above is true in the matching problem as $s = \sqrt{n}$. Here, we can say eventually

$$\delta_n \geq \frac{n^{m-1}}{\overline{\mathcal{D}}_n} \frac{q^2 s}{2} \geq \frac{q^2}{2n^{1/2}}$$

Hence, for all $m \geq 3$,

$$(\log n)^{3/2} = o(n^{\frac{m-2}{2}}) = o(\delta_n n^{\frac{m-1}{2}}).$$

We cannot satisfy the above condition for $m = 2$, and hence, we cannot conclude anything for $m = 2$. However, for $m \geq 3$, all conditions of Theorem 1 are satisfied and we can claim

$$|M_n| = O\left( \frac{(\log n)^2 n^{m+2}}{\frac{q^4}{4n} n^{2m}} \right) = O\left( \frac{(\log n)^2}{n^{m-3}} \right) \text{ a.s.},$$

which directly implies that for $m = 3$, $\frac{|M_n|}{n} \to 0$ a.s., and for $m > 3$, algorithm is consistent as $|M_n| \to 0$ a.s.