[Reviews · NeurIPS 2014]

Submitted by Assigned_Reviewer_6

This paper studies a planted partition model for random m-uniform
hypergraphs, and proves the consistency of a natural generalization of
spectral clustering. The hypergraph adjacency tensor is (mode-1)
flattened to a matrix, from which a normalized Laplacian matrix is
formed and the standard spectral partitioning is then applied. The
striking feature of the analysis is that the rate of convergence
improves as m increases, provided that the number of partitions is
small. Some experiments on both synthetic and application derived
data are reported, and the proposed method is shown to be relatively
effective, especially given its simplicity.

The model is well-motivated by applications in computer vision and
likely elsewhere. It is a very straightforward generalization of the
usual planted partition model / stochastic blockmodel. The proposed
method is nice in that is only based on matrix methods rather than
tensor decompositions (although this may also be a drawback), and the
analysis is relatively clean (at least the part I checked, namely
Lemma 4, which is the core of the analysis). For the most part (see
below), the paper clearly presents a nice result that seems to be
practically useful.

Higher-order affinities have been used in graph partitioning problems before, and they are not
at all mentioned in the present work. I think the author(s) should
review this literature (a few are listed below). I believe at least the
FSTTCS and COLT papers show other benefits of higher-order affinities.

On the Complexity of Random Satisfiability Problems with Planted Solutions
Vitaly Feldman, Will Perkins, Santosh Vempala
http://arxiv.org/abs/1311.4821

A tensor spectral approach to learning mixed membership community models
Anima Anandkumar, Rong Ge, Daniel Hsu, and Sham M. Kakade
COLT (2013)

Random Tensors and Planted Cliques
S. Charles Brubaker Santosh Vempala
RANDOM (2009)

A new approach to the planted clique problem
Alan Frieze and Ravi Kannan
FSTTCS (2008)

Max Cut for random graphs with a planted partition
Alexander Scott and Bela Bollobas
Combinatorics, Probability and Computing 13 (2004)

[I have read the authors' rebuttal.]
Summary: Overall, the paper gives a clearly presented and interesting
theoretical result on a generalization of the usual planted partition
model to uniform hypergraphs. I think it should be accepted provided
that the authors do a more thorough job in discussing related work.

Submitted by Assigned_Reviewer_22

In this paper the authors present theoretical guarantees for the accuracy of a spectral partitioning algorithm on uniform hypergraphs as well as experiments which support their claims. The proofs substantially build on prior work [5] regarding estimating misclustering.

This is an interesting paper and quite clearly written. The spectral clustering method presented and the theoretical guarantees appear novel. However, there are some questions in the minds of this reviewer which prevents declearing it a very good paper. Hopefully a revision that answers these issues and improves the presentation will follow.

1. Spectral partitioning for regular graphs that use eigenvectors of a Laplacian matrix formed from the adjacency matrix is well established. Applying a spectral technique for (uniform) hypergraphs got less attention, but there is prior art as also cited in the paper (especially [8]); these seem to all come from image processing backgrounds. There exists prior work from other areas that similarly forms a Laplacian (like) matrix derived from the hypergraph and uses eigenvectors of this matrix to perform partitioning or clustering. (Two examples from different fields: Multilevel Spectral Hypergraph Partitioning with Arbitrary Vertex Sizes J. Y. Zien, M. D. F. Schlag, and P. K. Chan in IEEE trans. on computer aided design of integrated circuits and systems vol 8, no 9 or Laplacian eigenvalues and partition problems in hypergraphs J. A. Rodriguez in Applied Mathematics Letters, volume 22, issue 6). One wonders how the proposed/discussed method differs from those other approaches.

The authors could properly place their work in the context of spectral techniques for hypergraphs in general and not just geometric groupings. It seems to this reviewer that the use of the flattened adjacency tensor is fairly new and therefore improvements to it and theoretical results might be novel. The authors could make it more clear what is novel and what is reproduced from [8] regarding the algorithm in their discussion.

2. The algorithm described in [2] slightly differs from algorithm 1 in that a normalization step is missing prior to the k-means clustering of the rows of X_n.

This reviewer asks the authors to discuss (in the paper and outside) what is the source of the difference between Algorithm 1 and that of [2]. Do normalization take place in the implementation?

3. In [5] on page 15 it is noted that the eigenvectors belonging to the largest absolute eigenvalues of L are (and should be) used in their framework. Since the proofs rely on [5], this reviewer asks the authors to discuss the form of the Lagrangian used and the subsequent eigenvector selection and how the misclustering estimates are affected. (See theorem 3.1 of [5] as well.)

4. Also in [5] on page 18 it is noted that their theorem 3.1 "depends on the true optimum of the k-means function", or more precisely a certain inequality holding. The authors of [5] repeated the k-means step with various random initializations as part of their simulations until the inequality did hold. This aspect should be addressed by the authors regarding their theoretical guarantees and experiments.

Some additional comments:

While the paper presents a spectral clustering algorithm it could be still beneficial to mention other state of the art hypergraph clustering approaches, for example Metis.

Throughout in the paper the nottaion is using n as subscript to remind the problem size. This only seems to make the nottaions slightly more complicated and this reviewer feels that the n subscript could be safely dropped. The notation can even be confusing, for example C_n is a kxk matrix and it is not denoting an nth iterate.

Line 104: "expected quantities of entries of A" - this does not sound right, this reviewer cannot parse "quantities of entries"

Throughout the paper there are mismatched articles like: "an hyperedge", "a m-uniform" etc.

Line 192: It seems that C_n is positive semidefinite and that could be mentioned.

Line 200: Some intuition could be given as to the various conditions.

Line 206: The statement |M| = O(T) almost surely may be confusing a bit at first for some readers. (It was for this reviewer.) This is interpreted as there exist a k such that |M| <= kT with probability 1. Is that correct?

Line 382: should be: "number of problems"

AFTER AUTHOR FEEDBACK:
I mentioned above the issue of the selection of the k eigenvectors of the Laplacian. I understand and thank the author's response regarding the Laplacian formulation. I am still a bit confused what formulation is used.

To elaborate: In [5] (on which this paper and the results build) the authors emphasize that the eigenvalues need to be ordered by absolute values. Now the standard definition of the Laplacian implies that it is positive semidefinite and so taking the abolute values appears unnecessary. However the authors of [5] are adamant; a quote from [5]:

"The above theorem suggests that before finding the largest
eigenvalues, you should first order them by absolute value. This allows for
large and negative eigenvalues. In fact, eigenvectors of L corresponding to
eigenvalues close to negative one (all eigenvalues of L are in [−1, 1]) discover
“heterophilic” structure in the network that can be useful for clustering."
(see page 16 in the discussion of theorem 3.1)

I repeat: this is mostly concerning since the submitted paper builds on and cites [5] heavily so this reviewer is puzzled: Did the authors of [5] make a mistake regarding the absolute value? Or did they use a different Laplacian than the submitted paper?

I do not generally hold the authors of the submitted paper responsible for the correctness of a cited work, but to the extent that no errors are propagated.
Summary: A promising paper that could be a good paper with a revision.

Submitted by Assigned_Reviewer_42

Summary:

This paper proposes an extension of the Stochastic Block Models for graphs to
m-uniform hyper-graphs, where an hyper-edge consists of exactly m nodes. Under
the proposed model, the authors design a spectral clustering algorithm that
operates on a flattened affinity tensor and prove its consistency under
certain conditions. They demonstrate that these conditions are satisfied
in two important computer vision problems, subspace clustering and point
set matching, which can be cast as clustering of m-uniform hyper-graphs
generated by special cases of the proposed model. Their consistency result
suggests that a larger size m of hyper-edges leads to faster convergence,
offering theoretical support for the efficacy of high-order matching and
multi-way affinity.

Comments:

Quality:

This paper is of decent quality. It connects useful applications in computer
vision with clustering on hyper-graphs generated by extensions of Stochastic
Block Models, and build on existing theoretical results on Stochastic Block
Models to provide guarantees for the proposed spectral clustering
algorithm on hyper-graphs. A nice technique they used is the
the flattened matrix form of high-order tensors, which allows them
to compute leading Eigenvectors efficiently and accurately, and to use some
existing results on spectral clustering.

Clarity:

The paper is clearly written and for the most part easy to follow.
There are a few places that can use extra clarification:
* Lines 166~168: Here the authors describe some existing related work, but it
is not clear how the proposed Algorithm 1 is different from them.
* Eq. (3): It looks like the smallest Eigenvalue of C_n controls how
identifiable the k groups are. Can the authors provide some more insight/high-level intuition on that?
What does C_n tell us? Why is its smallest Eigenvalue a critical quantity?
Of course these questions are all answered in the proofs, but some high-level insight would help to better convey the core idea of the analysis.
* Sec 5.2: How was the affinity computed? And what is the number of clusters?

Originality:

The proposed model for hyper-graphs seems to be a straightforward extension
of the Stochastic Block Model. Although Algorithm 1 is quite similar to the normalized spectral clustering algorithm, the use of the flattened tensor multiplied by its transpose to form a real, symmetric matrix seems novel and quite nice. The analysis associated with this modification also seems new compared with existing work on analyzing normalized spectral clustering, though the techniques involved are quite standard. Overall, this paper presents some original ideas and results, even
though they are not entirely surprising.

Significance:
Higher-order matching and multi-way affinity have had empirical success in
computer vision. The theoretical results in this paper explain to some degree
why it is the case. In this sense the paper is significant. However, it is not clear whether the proposed algorithm may help to advance the state of the art, as shown in the experiments. Its significance is limited in that regard.
Summary: A nice paper that connects useful applications in computer vision with
theoretical analysis of Stochastic Block Models and extensions. Although the
individual components of the algorithm and the analysis are fairly standard,
the authors combine them in a nice way to obtain some decent results.
Author Feedback
Author rebuttal: We thank the reviewers for their useful comments. We will revise the paper to incorporate the suggestions made by the reviewers. Detailed responses and clarifications are listed below.

Comments of Assigned_Reviewer_22:
1) Difference of Algorithm 1 from existing works:
We will add a discussion on other hypergraph partitioning approaches in the introduction in the final submission. Zien et al. (1999) considers partitioning general hypergraphs that are not uniform. The affinities cannot be expressed as symmetric matrices or tensors, and hence, it is difficult to analyze their approach using matrix perturbation results. Rodriguez (2009) focuses on uniform hypergraphs, but the definition of adjacency matrix is quite different and it does not use tensors.
Some additional references were cited by Assigned_Reviewer_6, and the difference is mentioned in the response to the comment.
In [8], only the leading eigenvector is required, which is computed using HOPM algorithm, and it is quantized to obtain a binary matrix representing matched points. We do not use HOPM and quantization. Also, in [8], the tensor is normalized, which we do not perform.

2) Difference with [2] in normalization:
We do not normalize the rows of the eigenvector matrix as done in [2]. This helps us to directly use the Davis-Kahan perturbation theorem [5, Theorem 2.1]. It is not clear how one would use this result if the row normalization is done.

3) Definition of L in [5]:
We are unable to follow this comment of the reviewer. In [5, Equation 1.3], L is defined as L = D^{-1/2}WD^{-1/2}. We use the same definition in Step 3 of Algorithm 1. Hence, the eigenvector selection and misclustering estimate also follow similarly.

4) True optimum of k-means:
In the theoretical results, we assume the true optimum of k-means as stated in Lemma 8. As mentioned in lines 315-318, we should run k-means multiple times. But, we empirically observed that a single run gives reasonably good results.

5) Use of subscript n:
Though C_n is kxk, it depends on Z_n which is a nxk matrix. We will add this clarification in the paper.

6) Intuition on conditions (line 200):
We thank the reviewer for this suggestion. We will include this in the revised version.

7) Meaning of |M| = O(T):
The reviewer's interpretation of the notation is correct.

8) Other textual corrections mentioned by the reviewer will be made in the final submission.

Comments of Assigned_Reviewer_42:
1) Clarifications for lines 166-168:
As mentioned in the text, this idea was proposed in [6]. Our contribution is in proposing the planted partition model for uniform hypergraphs, and proving the consistency of this spectral approach under the model. In [6], the problem is not viewed as partitioning random hypergraphs.
As suggested by the reviewer, we will explicitly state the difference among [6], [8] and Algorithm 1. In [6], W_n is approximately computed to reduce complexity, and the rows of eigenvector matrix are normalized before k-means clustering. We do not perform these steps to simplify the proof of the consistency result. In [8], only the leading eigenvector is required, which is computed using HOPM algorithm, and it is quantized to obtain a binary matrix representing matched points. We do not use HOPM and quantization. Also, in [8], the tensor is normalized, which we do not perform.

2) Insight into Eq. (3) and core idea of analysis:
We thank the reviewer for this suggestion. In the final submission, we will try to provide sufficient insight about the various conditions in the results.
As for C_n, it is obtained from flattening a k-dimensional tensor, that essentially contains the information in tensor B. We need the smallest eigenvalue of C_n to be large enough so that the partition (Z_n) can be determined from only the k leading eigenvectors of E[W_n]. This is a significant difference from standard spectral clustering, where it is much simpler to find Z_n from normalized expected affinity matrix, that is of rank k.

3) Clarifications for Sec 5.2:
The number of clusters is 2 (same as the number of independent motions), and the affinity is computed as the negative exponential of the error of fitting 4 points in a linear subspace of dimension 2 (similar to [6],[13]).

4) Comparison with state of the art:
The reviewer has mentioned that in the experiments, [13] and [8] perform better than Algorithm 1 in case of motion segmentation and matching, respectively. We would like to point out that both these state of the art approaches essentially use Algorithm 1, modified suitably to perform better in specific tasks. On the other hand, Algorithm 1 and our analysis (Theorem 1) are applicable in a much more general setting of finding any planted partition in a random uniform hypergraph, that need not be restricted to the settings of subspace clustering or matching.

Comments of Assigned_Reviewer_6:
1) Review of literature:
We thank the reviewer for this suggestion. We will add a discussion on the difference of Algorithm 1 with the works in COLT and FSTTCS in the final submission. These methods use a variational approach to define the eigenvectors, and do not construct an explicit eigen-decomposition of the tensor. An explicit decomposition is essential in our case for applying Davis-Kahan perturbation theorem.